# Assortment of Attention Heads: Accelerating Federated PEFT with Head Pruning and Strategic Client Selection

**Yeshwanth Venkatesha**                                          *yeshwanth.venkatesha@yale.edu*
*Department of Electrical Engineering*
*Yale University*

**Souvik Kundu**                                                      *souvikk.kundu@intel.com*
*Intel Labs*

**Priyadarshini Panda**                                               *priya.panda@yale.edu*
*Department of Electrical Engineering*
*Yale University*

**Reviewed on OpenReview:** `https://openreview.net/forum?id=WFpicZbAHe`

## Abstract

Parameter Efficient Fine-Tuning (PEFT) has become the de-facto approach in adapting Large Language Models (LLMs) for downstream tasks in Natural Language Processing. However, its adoption in privacy-preserving distributed learning frameworks, such as Federated Learning (FL), remains relatively limited. This is mainly due to challenges specific to FL, such as resource-constrained devices and diverse data distributions among clients. In this paper, we propose an efficient method to perform PEFT within the FL framework for Multi-Head Attention (MHA) based language models. We address the challenges through head pruning, a novel head-specific weighted aggregation mechanism, and a client selection strategy. Head pruning minimizes training complexity within the clients, guided by the importance score computed based on the confidence of the attention head. Weighted aggregation of heads ensures the global model captures crucial updates from diverse clients complementing our client selection strategy. We show results on the MultiNLI benchmark along with 20 Newsgroups, XL-Sum, and E2E NLG datasets. We use the MultiNLI dataset and T5-small model with LoRA as our PEFT method, attaining sparsity levels of up to 90%, resulting in a communication advantage of up to 1.8x and a reduction in training OPs of 3.9x while maintaining the accuracy drop under 2%.

## 1 Introduction

Large Language models (LLMs), such as GPT (Brown et al., 2020), T5 (Raffel et al., 2020), BART (Lewis et al., 2019) and BERT (Devlin et al., 2018), which are built on the Multi-Head Attention (MHA) based transformer architecture (Vaswani et al., 2017), have achieved exceptional outcomes in a wide range of Natural Language Processing (NLP) tasks. Moreover, they have begun to venture into other domains, including Computer Vision (CV) with models like VIT (Dosovitskiy et al., 2020), Stable Diffusion (Rombach et al., 2022), and LayoutLM (Xu et al., 2020), as well as Audio with models like Whisper (Radford et al., 2023) and XLS-R (Babu et al., 2021). Fine-tuning these models for downstream tasks such as text classification, summarization, and machine translation has proven to be highly effective. Specifically, a class of methods known as Parameter Efficient Fine-Tuning (PEFT) focuses on selectively updating a small subset of model parameters while keeping most of the pre-trained model unchanged (Hu et al., 2021; Li & Liang, 2021; Liu et al., 2021b). This approach significantly lowers the computational resources required to adapt large language models (LLMs) to specific downstream tasks. However, as the data sources become edge-focused and individuals grow more privacy-conscious, there is a shift towards privacy-preserving distributed learning

paradigms, such as Federated Learning (FL) (McMahan et al., 2017) for several applications (Hard et al., 2018; Chen et al., 2019; Liu et al., 2021a; Babakniya et al., 2023b). Naturally we would need to do federated learning for language model fine tuning.

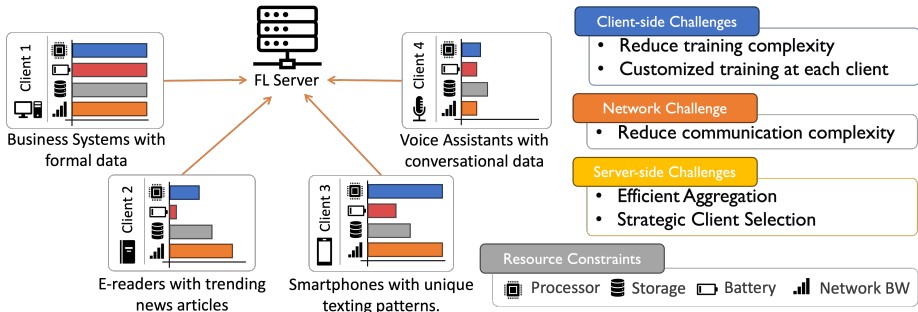

Figure 1: Within a federated environment, devices often face resource limitations and exhibit variability in both data and available resources. Numerous challenges arise at each stage when finetuning a language model under these conditions.

As illustrated in Fig. 1, consider a scenario of FL involving distributed sources of text data. Each source exhibits unique text patterns, encompassing diverse text styles. For instance, business systems may contain formal documents, while voice assistants may comprise casual conversational data. Furthermore, each client is constrained by individual device limitations, such as battery life, processing power, storage, and connectivity. The process of training a model in such an environment poses challenges on multiple fronts. On the client side, constraints arise due to limited and often diverse resources across clients. On the server side, efficient aggregation and client selection strategies become imperative to handle client heterogeneity and accelerate convergence. Additionally, the communication cost between the client and the server constitutes an additional overhead.

Transformer models with MHA are shown to effectively capture intricate textual details, as specific heads encode syntax, rare words, and positional nuances. Additionally, it has been demonstrated that several heads are redundant and can be pruned without sacrificing the accuracy (Voita et al., 2019; Li et al., 2021; Behnke & Heafield, 2020; Shim et al., 2021). In this work, we investigate harnessing the inherent structure offered by multiple heads in MHA mechanism to devise an efficient FL PEFT method. Our goal is to minimize the training and communication complexity for individual clients while facilitating a cohesive aggregation of models independently trained on their private data. To this end, we first implement head pruning to reduce the training and communication complexity at the client level. The pruning decision is guided by the importance score specific to each client's heads. The importance score works as a fingerprint of the data present in the clients. Next, we perform weighted averaging of the retained heads, ensuring that the server accentuates the information from unique data distributions from all participating clients.

Further, in typical large-scale FL systems, the sheer number of devices makes it essential to select a subset of devices for each FL round. This client selection directly impacts the performance of the system, making a tailored approach necessary. We propose a client selection strategy based on the difference in clients' local loss and global model loss prioritizing the updates from clients with maximal impact. The weighting mechanism of attention heads proves especially beneficial when a limited number of clients are chosen for training. Put simply, our approach addresses the training and management of multiple attention heads trained from distributed data sources, hence the title "Assortment of Attention Heads". Our contributions are summarized as follows.

- To minimize training and communication complexity at the clients, we implement a head pruning strategy using head importance scores determined by the maximum attention scores generated by each head.
- We introduce an aggregation mechanism in which updates from each head are weighted according to their importance, allowing for the collection of the most crucial updates to the global model. This

effectively mitigates the performance drop caused by sparsity, thereby enhancing the convergence rate.

- We propose a client selection strategy based on the loss difference between the global model and the client model to further improve performance.

- We demonstrate the efficacy of our approach on two text classification datasets, MultiNLI and 20 Newsgroups, as well as on more demanding summarization task using the XL-Sum dataset and generation task using E2E NLG dataset. On MultiNLI dataset, we achieve 90% head pruning translating to 1.8 times and 3.9 times reduction in communication complexity and training operations (OPs) respectively compared to training the full network with standard FedAvg. Additionally, we achieve faster convergence compared to random client selection at similar levels of sparsity.

## 2 Background and Related Work

**Muti-Head Attention:** The standard attention mechanism in transformers (Vaswani et al., 2017) is defined as:

$$\text{Attn}(Q, K, V) = \text{softmax}\left(\frac{QK^T}{\sqrt{d_k}}\right) \cdot V \tag{1}$$

where Q, K, and V correspond to Query, Key, and Value matrices respectively. In practice, the transformer model uses multiple attention heads to capture different aspects of the relationships between words. Let $h$ denote the index of the attention head. Each attention head computes a different set of query, key, and value vectors. The outputs of all attention heads are then concatenated and linearly transformed to obtain the final output. The multi-head attention (MHA) operation is defined as follows:

$$\text{MHA}(Q, K, V) = \text{Concat}(\text{head}_1, \ldots, \text{head}_h) \cdot W_O \tag{2}$$

where

$$\text{head}_h = \text{Attn}(Q \cdot W_{Qh}, K \cdot W_{Kh}, V \cdot W_{Vh}) \tag{3}$$

Here, $W_{Qh}$, $W_{Kh}$, $W_{Vh}$, and $W_O$ are learnable weights corresponding to each attention head.

**Parameter Efficient Fine-Tuning:** Transfer learning has been a pivotal approach in leveraging pre-trained models for various natural language processing tasks. Researchers have explored techniques for fine-tuning pre-trained models on specific downstream tasks to achieve improved performance with fewer computational resources. The umbrella term for this is Parameter Efficient Fine Tuning (PEFT). Adapters (Houlsby et al., 2019), LoRA (Hu et al., 2021), AdaLoRA (Zhang et al., 2023a), IA3 (Liu et al., 2022a), BitFit (Zaken et al., 2021), and Ladder Side Tuning (LST) (Sung et al., 2022) represent some of the widely adopted techniques in this domain. Prefix tuning strategies involve adjusting the model's prompt or prefix to tailor its behavior for specific tasks (Li & Liang, 2021; Liu et al., 2021b; 2023; Lester et al., 2021). Our approach is not limited to a specific PEFT strategy; rather it can be applied to any PEFT strategy as long as its trainable parameters can be associated with the heads of the MHA mechanism.

**Federated Learning with Transformers:** Previous works have explored federated learning methods specifically catered for transformer models. There are several works that benchmark NLP tasks that primarily rely on transformer models within the federated learning environment, as evidenced by (Hilmkil et al., 2021; Lin et al., 2021). In an alternative research direction, personalized federated learning approaches using transformer architectures have been explored for customizing models to individual clients (Li et al., 2023; Sun et al., 2023; Kim et al., 2023). Simultaneously, in computer vision, works have compared CNNs and transformer models for vision tasks in federated learning (Qu et al., 2022; Chen et al., 2022; Zuo et al., 2022). Specifically addressing PEFT in federated learning, (Hyeon-Woo et al., 2021; Babakniya et al., 2023a; Yan et al., 2024) leverage the LoRA framework for efficiency, and (Zhao et al., 2022) studies prompt tuning effects. Various studies benchmark diverse PEFT methods in federated learning, covering privacy and resource constraints (Zhuang et al., 2023; Zhang et al., 2023b; Sun et al., 2022). While prior works adapt transformers to the federated learning paradigm, they have not exploited the natural structure of the multi-head attention mechanism for training and communication efficiency.

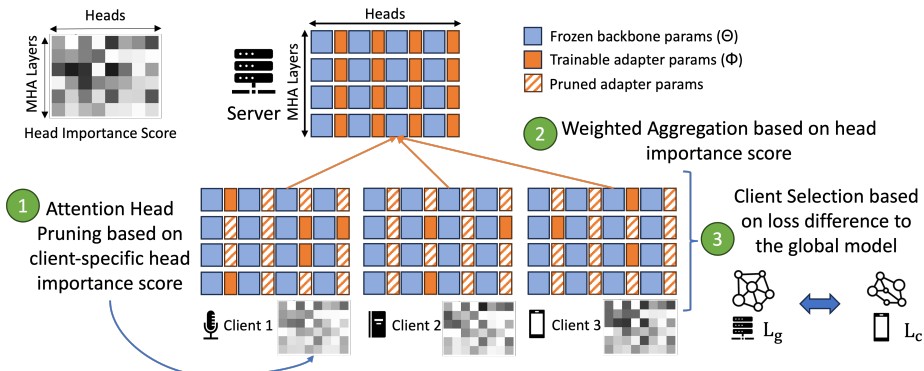

Figure 2: An overview of our strategy encompassing attention head pruning guided by importance scores and weighted aggregation, alongside client selection determined by the difference in loss between the global server model and the local client model.

**Pruning in Transformers:** Pruning is a well-known process in machine learning where the redundant parts of the model are removed to achieve efficient processing (Sietsma & Dow, 1988; LeCun et al., 1989; Han et al., 2015; Blalock et al., 2020). Although initially popularized in the vision domain, adaptations of pruning techniques have been applied to transformer models as well (Mao et al., 2021; Lagunas et al., 2021). The concept of head pruning, as presented in the "Story of heads" (Voita et al., 2019), demonstrates that a significant number of heads can be pruned without sacrificing performance, a notion extended by subsequent works such as (Held & Yang, 2022; Wang et al., 2021). The authors of (Gordon et al., 2020) study the effect of pruning the backbone model on fine-tuning. We take inspiration from previous work and adapt to federated settings.

**Client Selection in FL:** Federated learning typically assumes unbiased client selection, randomly choosing a subset for each round. However, biasing this selection can potentially improve convergence rates and mitigate non-IID data concerns (Cho et al., 2022; Zhang et al., 2021; Fu et al., 2023). Some works adopt a systems approach, selecting clients based on resource availability, particularly in IoT applications (Nishio & Yonetani, 2019; Xu & Wang, 2020; AbdulRahman et al., 2020). The client selection algorithm in (Huang et al., 2020) incorporates fairness considerations. Prior approaches often rely on pre-communication local training, or resource monitoring. In contrast, our method selects clients based on the most recent loss value of their local models, prioritizing clients whose updates are expected to provide the highest impact on the global model. This adaptive, loss-based client selection, when combined with attention-head pruning and importance-weighted aggregation, constitutes a novel system-level contribution that improves convergence efficiency while remaining computationally lightweight.

## 3 Methodology

Consider a PEFT setup with model $M = \{\Theta, \Phi\}$ where $\Theta$ is the frozen backbone parameters and $\Phi$ is the set of trainable parameters (See Fig. 2). A federated learning system consists of a server and $C$ clients indexed with $[C]$, where each client $c$ contains a local dataset $D_c$. The federated learning process spans over $R$ rounds of communication. In round $r$, the server selects a subset of clients $S^{(r)}$ ($\ll C$), and broadcasts the trainable parameters $\Phi^{(r-1)}$ for local training. The objective of each client is to minimize the local loss $\mathcal{L}_c$ while updating only the local trainable parameters $\Phi_c$:

$$\underset{\Phi_c}{\arg\min}\, \mathcal{L}_c(\Theta, \Phi_c). \tag{4}$$

After the clients complete their local training over a certain number of epochs, the server gathers their model updates and integrates them into the server model as follows:

$$\Phi^{(r)} = \Phi^{(r-1)} + \eta \frac{\sum_{c \in S^{(r)}} |D_c| \cdot \Delta\Phi_c^{(r)}}{\sum_{c \in S^{(r)}} |D_c|} \tag{5}$$

Here, $\Delta\Phi_c^{(r)} = \Phi_c^{(r)} - \Phi^{(r-1)}$ is the model update from client $c$ at round $r$, and $\eta$ is the server learning rate controlling global model changes each round. The model updates are weighted by their respective data sample count.

We identify the inefficiencies in this framework by asking the following key questions:

1. Is it necessary to train all the parameters across all clients? Can we achieve comparable performance by selectively training a subset of $\Phi_c$?
2. How can we modify the aggregation mechanism with custom weight to each parameter to maximize the collective knowledge aggregated from different clients?
3. What strategy can we employ to optimize the selection of $S^{(r)}$ for maximizing the performance?

In the following sections, we tackle these concerns through our proposed strategies. As illustrated in Fig. 2, we tackle the problem with three key methodologies. First, we reduce the training and communication demands of each client by performing a head pruning which we describe in Section 3.1. Next, we propose a novel weighted aggregation technique to maximize the knowledge transfer from pruned models (refer to Section 3.2). Finally, we introduce a client selection approach to improve the convergence rate (refer to Section 3.3).

## 3.1 Head Pruning

Assuming a multi-head attention model with $H$ heads, designate a subset of trainable parameters $P \subseteq \Phi$ such that $P = \{p_1, p_2, ..., p_H\}$, each associated with a head indexed by $[ht]$. We prune away a subset of these parameters to increase the efficiency of the local training as well as the communication cost (see Fig. 2). To determine which parameters to prune, we compute the importance of each attention head at a given client by averaging the maximum attention scores (excluding the end-of-sentence symbol) across the client's local dataset (Voita et al., 2019). This score reflects the "confidence" of an attention head in its predictions, which we use as a proxy for its importance. The resulting importance score also serves as a fingerprint of the client's data distribution. Formally, the importance score of head h at client c is defined as:

$$\alpha_{hc} = \frac{1}{|D_c|} \sum_{i \in D_c} \max_t \left( (q_h^{(i)}(t))^\top \cdot k_h^{(i)}(t) \right), \tag{6}$$

where $q_h^{(i)}(t)$ and $k_h^{(i)}(t)$ denote the query and key vectors of head h for token t in the i-th data sample. The max is taken over the token positions in the sequence, and the final score is averaged across all samples in the client dataset $D_c$. We then prune the heads whose importance is below a certain threshold calculated according to the desired level of sparsity as:

$$\alpha_{hc} = \begin{cases} \alpha_{hc} & \text{if } \alpha_{hc} \geq \text{threshold} \\ 0 & \text{otherwise} \end{cases} \tag{7}$$

Note that the values of $P$ corresponding to the pruned values of $\alpha$ need not be communicated to the server thereby saving a significant amount of communication cost.

We opt for Low-Rank Adapters (LoRA) (Hu et al., 2021) as our chosen PEFT approach. In LoRA, the gradient of each weight matrix in the MHA layer is decomposed into two low-rank matrices, denoted as $\Delta W = BA$. Our updates are then applied exclusively to these parameters and the designated task head, denoted as $T$. Consequently, our trainable parameter set is denoted as $\Phi = \{A, B, T\}$.

For an input token $x$ with a dimensionality of $d$, the matrix $A$ maps it to a lower dimension $d'$, and this lower-dimensional representation is then mapped back to the original dimensionality of $d$ by matrix $B$. Since multi-head attention operates along the $d$ dimension, we can distribute only the matrix $B$ across the attention heads. Thus, in this context, the set $P = \{B\}$. Note that while we illustrate the application with LoRA, our approach can be extended to any PEFT method where trainable parameters can be mapped to specific attention heads.

## 3.2 Weighted Aggregation

At round $r$, let the model update corresponding to head $h$ from each client $c$ be denoted by $\Delta p_{hc}^{(r)}$. We weight $\Delta p_{hc}^{(r)}$ by their importance score $\alpha_{hc}^{(r)}$ to update the corresponding parameters in the global model $p_{hg}^{(r)}$ as follows:

$$p_{hg}^{(r)} = p_{hg}^{(r-1)} + \eta \frac{\sum_{c \in S^{(r)}} \alpha_{hc}^{(r)} \cdot \Delta p_{hc}^{(r)}}{\sum_{c \in S^{(r)}} \alpha_{hc}^{(r)} + \epsilon} \tag{8}$$

$\epsilon$ is a small constant added for numerical stability and $\eta$ is the server learning rate controlling the rate of global model updates. The weighting of updates based on the importance of each head for a specific client ensures that the global model extracts the maximum benefit from the individual client contributions. The remaining trainable parameters that cannot be assigned to an MHA head (the set $\Phi - P$) are updated following Equation 5.

## 3.3 Client Selection

To determine participating clients $S^{(r)}$, we select the top $K$ clients with the highest difference in loss compared to the global model. Using the server loss $\mathcal{L}_g^{(r-1)}$ (the loss of the global model at the server) and each client's loss from their last communication $\mathcal{L}_c^{(r')}$, we calculate the difference and choose clients with the maximum disparity:

$$S^{(r)} = \underset{1,\ldots,K}{\arg\max}(\mathcal{L}_c^{(r')} - \mathcal{L}_g^{(r-1)}) \tag{9}$$

The rationale behind this selection criterion is that as this difference increases, it indicates that the client is falling behind the global model. Consequently, training updates from such clients are expected to contribute more significant improvements when incorporated into the global model. We initiate all clients with high loss values to promote initial engagement from clients that have not participated previously. We summarize the overall flow of our method in algorithm 1.

---

**Algorithm 1:** Federated PEFT Algorithm

**Input:** Model $M = \{\Theta, \Phi\}$, $C$ clients, $R$ rounds, learning rate $\eta$

**for** $r = 1$ *to* $R$ **do**

    Select the set of participating clients $S^{(r)}$ (Eqn. 9);

    Broadcast $\Phi^{(r-1)}$ to $S^{(r)}$;

    **for** *each client* $c \in S^{(r)}$ **do**

        Perform local training (Eqn. 4);

        Compute model updates;

        Compute importance score $\alpha_c$;

        Communicates sparse parameters $P_c$ and head importance scores $\alpha_c$;

    Server updates global model (Eqn. 8);

---

# 4 Experiments

We consider the popular text classification datasets MultiNLI (Williams et al., 2018) and 20 Newsgroup (Lang, 1995) and two generation datasets, XL-Sum (Hasan et al., 2021) and E2E NLG (Novikova et al., 2017). MultiNLI dataset contains over 400k sentence pairs mapped to 3 classes—entailment, contradiction, and neutral. 20 Newsgroup is a collection of nearly 20k documents spanning 20 different newsgroups ranging from technology and science to politics and sports. We take the English subset of XL-Sum dataset which contains over 330k samples of annotated article-summary pairs. E2E NLG is a data-to-text generation task with 42k samples.

We uniformly distribute the training data among clients for each dataset and conduct evaluations on the held-out validation set. We adopt publicly available splits for training and validation data in XL-Sum and 20

| Method | CS | Acc. | Conv Rate | Sparsity | OPs | Comm. Complexity |
|---|---|---|---|---|---|---|
| B1. FedAvg | Rand | $74.03 \pm 0.25$ % | 5 | 0% | 33.07G | 3.38MB |
| | Loss | $78.76 \pm 0.45$% | 5 | | | |
| B2. FedAvg + Rand Pruning | Rand | $71.39 \pm 0.45$ % | 17 | 90% | 8.47G | 1.86MB |
| | Loss | $76.87 \pm 0.39$ % | 17 | | | |
| Ours | Rand | $72.76 \pm 0.38$% | 9 | 90% | 8.47G | 1.86MB |
| | Loss | $76.94 \pm 0.34$ % | 11 | | | |

Table 1: Ablation study on the client selection (CS) and head pruning strategies on MultiNLI dataset with 100/2 clients. For each of the baselines we show results for both random client selection (Rand) and strategic loss-difference based clients selection (Loss). In addition to accuracy, we also provide information on the convergence rate, quantified by the number of rounds needed to reach a 70% accuracy threshold.

Newsgroup. In the case of MultiNLI, we merge both matched and mismatched validation sets into a unified validation set. Across all experiments, a batch size of 80 and a local learning rate of 5e-4 are utilized. We employ one local epoch and conduct 100 communication rounds. All the experiments are performed on a machine with one A100 GPU with 80GB GPU memory and an 8-core CPU with 16GB memory per core.

In addition to accuracy, we measure the computation and communication complexity as follows:

- **Computation Complexity (OPs)**: Specifically, we account for the forward and backward computation of Q, K, V matrices, attention layers, and the subsequent fully connected layers. We use the maximum token length of $T = 512$ across all our experiments. For simplicity, we omit the peripheral layers such as tokenizer, positional encoding, and final classification/language model head since they are common across all the methods. Details are provided in A.
- **Communication Complexity**: For communication complexity, we calculate the total number of trainable parameters in each client since only these needs to be communicated to the server each round.

We employ the MultiNLI dataset for a comprehensive analysis, evaluating the influence of our client selection approach and the significance of head pruning based on importance scores. We assess their effects on both the accuracy and convergence rate of the model, measured by the number of rounds required to achieve 70% accuracy. We establish the following baselines to compare the effectiveness of our design.

- B1: Standard federated averaging without head pruning (FedAvg).
- B2: Federated averaging with randomly selected heads for pruning (FedAvg + Random Head Pruning).

In our method we employ a head pruning of 90% based on the head importance score (Eqn. 6). For each of the methods B1, B2 and ours we show results for both random client selection and loss based strategic client selection in Table 1. We summarize the observations as follows:

- Loss difference-based client selection directly influences the accuracy by approximately 5%, regardless of the chosen head pruning strategy, as shown in the comparison between two consecutive rows.
- Client selection does not impact the convergence rate independently, as indicated by the consistent number of rounds required to reach 70% accuracy in both B1 (5 rounds) and B2 (17 rounds).
- Head pruning plays a significant role: transitioning from no sparsity in B1 to 90% sparsity with random pruning in B2 results in a noticeable decrease in both accuracy ($\sim$2%) and results in slower convergence (11 rounds).
- Importance score based head pruning strategy alone enhances convergence rate (17 rounds in B2 to 9 rounds in ours) and improves accuracy by $>$1% compared to random pruning (B2).

| Dataset | # Clients | FedAvg (B1 + Random CS) | | | Ours | | |
|---|---|---|---|---|---|---|---|
| | | Comm. | OPs | Acc. | Comm. | OPs | Acc. |
| MultiNLI | 10/2 | 3.38MB | 33.07G | 78.34 ± 0.23 % | 1.86MB | 8.47G | 79.13± 0.43% |
| | 100/2 | 3.38MB | 33.07G | 74.03 ± 0.25% | 1.86MB | 8.47G | 76.94± 0.34% |
| | 1000/20 | 3.38MB | 33.07G | 67.35± 0.63% | 1.86MB | 8.47G | 67.05 ± 0.66% |
| Newsgroup | 10/2 | 5.06MB | 228.90G | 74.53± 0.43% | 2.78MB | 78.34G | 77.10 ± 0.68% |
| | 100/2 | 5.06MB | 228.90G | 72.01 ± 0.56% | 2.78MB | 78.34G | 77.59± 0.63% |
| | 1000/20 | 5.06MB | 228.90G | 67.89± 1.03% | 2.78MB | 78.34G | 75.56± 0.83% |

| Dataset | # Clients | FedAvg (B1 + Random CS) | | | Ours | | |
|---|---|---|---|---|---|---|---|
| | | Comm. | OPs | R1/R2/RL | Comm. | OPs | R1/R2/RL |
| XL Sum | 10/2 | 3.38MB | 33.07G | 0.34/0.12/0.33 | 1.86MB | 8.47G | 0.35/0.12/0.32 |
| | 100/2 | 3.38MB | 33.07G | 0.33/0.12/0.32 | 1.86MB | 8.47G | 0.34/0.12/0.33 |
| | 1000/20 | 3.38MB | 33.07G | 0.33/0.12/0.31 | 1.86MB | 8.47G | 0.34/0.12/0.31 |
| E2E NLG | 10/2 | 3.38MB | 33.07G | 0.56/0.29/0.50 | 1.86MB | 8.47G | 0.55/0.32/0.51 |
| | 100/2 | 3.38MB | 33.07G | 0.55/0.29/0.50 | 1.86MB | 8.47G | 0.51/0.26/0.42 |
| | 1000/20 | 3.38MB | 33.07G | 0.48/0.13/0.41 | 1.86MB | 8.47G | 0.47/0.13/0.42 |

Table 2: Results of our method across four datasets over different number of clients. For instance, in the configuration "100/2", a total of 100 clients exist, with 2 participating in each training round. We provide accuracy metrics for classification tasks and ROUGE metrics (ROUGE-1/ROUGE-2/ROUGE-L) for the generation tasks. We show the baseline of using standard FedAvg without any sparsity and random client selection and compare it to our method with 90% sparsity and loss-based client selection.

- Combining head importance-based pruning with loss-based client selection yields higher accuracy and convergence rate compared to FedAvg with random pruning (B2) and achieves similar accuracy to FedAvg without pruning (B1).
- We achieve a 3.9-fold reduction in computation complexity, as measured by OPs, and a 1.8-fold enhancement in communication complexity compared to dense baseline (B1).

To delve deeper, we plot the validation accuracy during the initial 30 rounds of training in Fig. 3. The baseline B1 (FedAvg) with loss based client selection is the best performing among our experiments with higher accuracy as well as faster convergence. On the other extreme, baseline B2 (FedAvg + random head pruning) with random client selection serves as the lower limit. We see that importance-based head pruning and weighted aggregation significantly improves the convergence rate, illustrating that our strategy enables positive collaboration among clients in the early stages of training. However, as training progresses to the later stages where targeted model updates are crucial, the client selection strategy becomes more pivotal. Thus, both strategies complement each other, enabling training approaches that achieve performance levels close to the best-performing baseline scenario of no sparsity (B1) with loss-based strategic client selection.

We showcase the effectiveness of our methodology across four datasets, while considering diverse client configurations. We present accuracy as the primary performance measure for classification tasks, and ROUGE (ROUGE-1/ROUGE-2/ROUGE-L) scores for summarization and summarization/generation tasks. To establish a baseline for comparison, we report the accuracy of standard FedAvg (B1 + Random CS). Our analysis encompasses scenarios involving 10, 100, and 1000 clients, each with a 2% participation rate, denoted as 100/2 and 1000/20 configurations. In the case of 10 clients, we consider the minimal participation of 2 clients. Across MultiNLI, XL-Sum, and E2E NLG tasks, we utilize T5-small as the backbone, while employing BART-base for the 20 Newsgroup task. LoRA serves as the PEFT method across all datasets. We apply a head pruning of 90% according to the calculated head importance scores. The aggregation process utilizes these scores and pruned gradients to update the global model. Our method exhibits effectiveness, particularly in scenarios with a high number of clients. Importantly, its applicability extends beyond conventional text

| Method | Training Time | FedAvg (B1 + Random CS) | | | Ours | | | |
|--------|---------------|-------|-----|------|---------|-------|-----|------|
| | | Comm. | OPs | Acc. | Max Pr% | Comm. | OPs | Acc. |
| FFT | 47s | 41.96MB | 41.07G | 75.60± 0.63% | 90% | 23.08MB | 10.63G | 78.24± 0.59% |
| LoRA (r = 16) | 38s | 3.38MB | 33.07G | 74.03 ± 0.25 % | 90% | 1.86MB | 8.47G | 76.94± 0.34% |
| LoRA (r = 8) | 38s | 1.69MB | 32.38G | 75.30± 0.24% | 90% | 0.93MB | 8.00G | 76.67± 0.29% |
| LoRA (r = 4) | 38s | 0.84MB | 32.05G | 75.22± 0.45% | 90% | 0.46MB | 7.76G | 76.07± 0.43% |
| IA3 | 36s | 168KB | 38.70G | 59.60± 1.12% | 50% | 84KB | 21.76G | 60.14± 1.03% |
| Prompt Tuning | 32s | 743KB | 22.81G | 48.46± 0.43% | 90% | 74KB | 6.47G | 53.93± 0.43% |
| P-Tuning | 32s | 1.80MB | 23.80G | 57.28± 0.85% | 90% | 0.18MB | 6.73G | 60.83± 0.59% |

Table 3: Effect of PEFT method. We compare several PEFT methods along with the full fine-tuning (FFT) baseline in the standard MultiNLI setting with 100/2 clients. We report training time per epoch, maximum sparsity before accuracy drops to random guessing levels (Max Pr%), and communication cost (Comm.), MAC operations (OPs), alongside accuracy measurements (Acc.).

classification tasks, proving its effectiveness even in challenging tasks such as summarization and natural language generation.

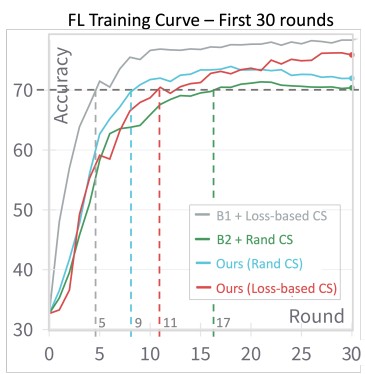

Figure 3: Training curves over the first 30 rounds illustrating the impact of importance score-based head pruning and weighted aggregation. Baselines B1 and B2 are shown along with our method without strategic client selection. Pruning and aggregation accelerate early convergence, while client selection improves later-stage performance.

**Effect of Sparsity:** Here, we investigate the impact of head pruning on the performance of our method. In Fig. 4 we show the accuracy, number of OPs, and communication cost measured by the number of trained parameters sent by each client on an average per round. We vary sparsity across a range of sparsity levels, from 0% to 95%. Accuracy experiences only a minimal decline up to 90% sparsity; however, beyond 95%, the accuracy diminishes to a level equivalent to random guessing. At 90% sparsity, we observe 1.8x improvement in communication cost and 3.9x lower MAC operations with less than a 2% drop in accuracy.

**Effect of PEFT Method:** In this study, we analyze the impact of various fine-tuning methods on our proposed approach. Specifically, we examine the outcomes of full fine-tuning (FFT), where all parameters of the backbone model are fine-tuned, alongside several established PEFT methods including IA3 (Liu et al., 2022a), P-Tuning (Liu et al., 2022b), and Prompt Tuning (Lester et al., 2021). Additionally, we present results for different ranks in LoRA. The findings are summarized in Table 3, which includes accuracy and system metrics training time, communication cost, and the OPs count per local round. We report the maximum sparsity achieved with our method before the model shows diverging behaviour. We include the baseline of standard FedAvg (B1 + Random Client Selection) for comparison. LoRA improves over full fine-tuning (FFT) in terms of OPs by 20% and >12x in terms of communication cost with minimal accuracy drop. In our method, decreasing the rank within LoRA leads to a steady decline in accuracy. While the improvement in computation complexity is marginal there is a proportional enhancement in communication complexity. This parameter can be tuned based on the specific application requirements.

With IA3, Prompt tuning, and P-Tuning while we can achieve a significant reduction in communication cost, they are not competitive in the final accuracy compared to LoRA. Between P-tuning and prompt tuning, P-tuning demonstrates better performance owing to trainable tokens at every layer. However, it comes with the trade-off of increased communication complexity. Additionally, while IA3 is more efficient than LoRA with ∼40x less communication cost, it is also extremely sensitive to sparsity shown by the fact that we can only go up to 50%. Note that the training time can be further improved with a custom implementation of pruned model training.

| Model Type | Model Name | Training | FedAvg (B1 + Random CS) | | | Ours | | |
|---|---|---|---|---|---|---|---|---|
| | | Time | Comm. | OPs | Acc. | Comm. | OPs | Acc. |
| EncDec | T5-Small | 38s | 3.38MB | 33.07G | 74.03± 0.25% | 1.86MB | 8.47G | 76.94± 0.34% |
| EncDec | T5-Base | 166s | 10.13MB | 128.19G | 82.52± 1.19% | 5.57MB | 34.83G | 79.03± 0.69% |
| EncDec | BART-Base | 82s | 5.06MB | 209.58G | 66.80± 0.53% | 2.78MB | 59.01G | 78.77± 0.54% |
| Enc-only | DistilBERT | 33s | 6.22MB | 42.73G | 74.00± 0.58% | 3.42MB | 11.61G | 75.11± 1.03% |
| Enc-only | RoBERTa | 64s | 7.89MB | 21.89G | 77.68± 0.68% | 4.34MB | 5.63G | 82.24± 0.43% |
| Dec-only | GPT-2 Small | 10s | 3.94MB | 42.73G | 0.67/0.42/0.64 | 2.16MB | 11.61G | 0.69/0.44/0.66 |

Table 4: Effect of different backbone architecture. Besides our usual experimental configuration using T5-small, we incorporate a larger model represented by T5-base. Moreover, for comparative analysis, we incorporate both the base variant of the BART model. We include two encoder-only models in DistilBERT and RoBERTa in addition to a decoder-only model GPT-2. Our comparisons encompass training time per epoch, communication cost, and the maximum sparsity achieved without encountering model divergence. We use the standard configuration of MultiNLI dataset with 100/2 clients and LoRA as our PEFT method.

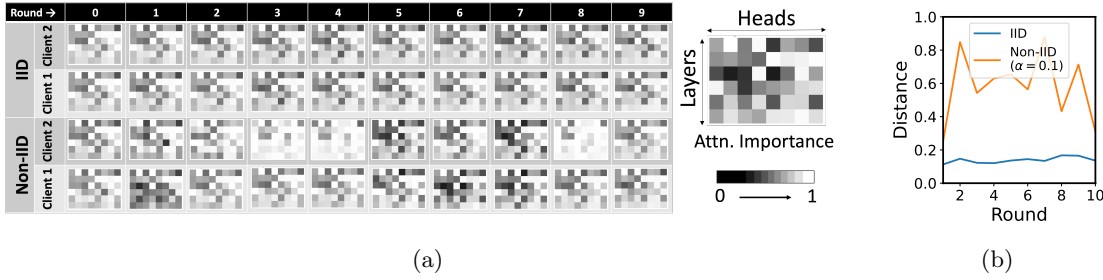

(a)        (b)

Figure 5: Assessment of head importance scores under IID and Non-IID settings. (a) Head importance visualization for encoder-decoder cross-attention across two clients per round, during the first 10 rounds of training. (b) Quantitative comparison using Euclidean distance between the two clients' importance scores in each round over the first 10 rounds.

**Effect of Backbone Model:** We also measure the influence of substituting the backbone model on the overall system metrics in Table 4. Alongside the standard T5-small encoder-decoder model utilized in our experiments, we incorporate a larger T5-base model and BART (Lewis et al., 2019). Furthermore, we introduce two encoder-only models, DistilBERT (Sanh et al., 2019) and RoBERTa (Liu et al., 2019) for the classification task MultiNLI. Finally, we include a decoder-only model GPT-2 (Radford et al., 2019) to show the performance on E2E NLG task. We employ the case of 100/2 clients and measure the same parameters—training time per epoch, maximum sparsity achieved before the model diverges, communication cost, OPs, and accuracy. We find our approach consistently outperforms the baseline across all metrics. Employing a larger model certainly improves performance, but it comes with trade-offs. The training time increases by more than 4x, and the communication cost and OPs count triple when transitioning from T5-small to T5-base. Among encoder only models, RoBERTa provides better accuracy at a lower compute complexity in terms of OPs. However, the communication complexity is higher than that of T5-small.

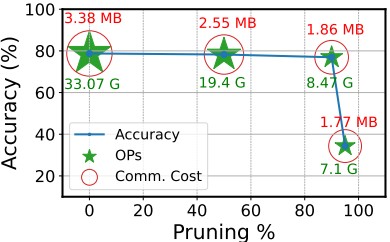

Figure 4: Effect of pruning on accuracy, communication cost and OPs.

**Effect of Non-IID data:** We examine the impact of non-IID data in this section. We regulate the *Non-IIDness* by adjusting the $\alpha$ parameter in the Dirichlet distribution. Employing our standard experimental setup with MultiNLI, 100/2 clients, and LoRA as the PEFT method, we present the results in Table 5. Notably, even under extreme non-IID data conditions ($\alpha = 0.1$), our method exhibits robustness, as we observe no performance degradation. In Fig. 5, we offer a qualitative and quantitative analysis of head

importance scores for IID vs Non-IID scenarios. These scores are computed for each selected client over the initial 10 rounds of training. While both clients demonstrate similar scores in the case of IID (top 2 rows), a noticeable divergence appears in the case of non-IID, as illustrated in Fig. 5a. To quantify this difference, we plot the Euclidean distance between the scores for both scenarios in Fig. 5b. The distance between the clients is low and nearly constant in the case of IID whereas in the case of non-IID the distance is significantly higher and has larger variation across rounds.

## 5 Conclusion and Discussion

In this paper, we present our method to improve the training and communication complexity of training PEFT in an FL environment. We accomplish this by capitalizing on the inherent structure within the multi-head mechanism of transformer models, employing a combination of attention-head pruning and strategic client selection. The empirical results demonstrate the efficacy of our approach across various datasets such as MultiNLI, 20 Newsgroup, XL-Sum, and E2E NLG. We conduct additional experiments to validate the effectiveness of our approach under diverse scenarios, encompassing non-IID data settings, different PEFT methods, and backbone models.

| Dirichlet Param $\alpha$ | FedAvg | Ours |
|:---:|:---:|:---:|
| IID | 74.03± 0.25% | 76.94± 0.34% |
| 2.0 | 74.61± 0.32% | 77.70± 0.26% |
| 0.5 | 74.41± 0.13% | 76.91± 0.29% |
| 0.1 | 74.45± 0.22% | 77.65± 0.34% |

Table 5: Effect of Non-IID data. We vary the parameter $\alpha$ of Dirichlet distribution to increase the degree of *non-IIDness* of the data. The results shown are for the standard case of LoRA on MultiNLI with 100/2 clients.

While our exploration is limited to head pruning, we acknowledge numerous other avenues for enhancing model efficiency, such as token pruning, individual weight pruning, quantization, linear attention, and more. The challenges associated with integrating these methods into a federated environment remain an ongoing and open problem that merits further exploration. Our research is confined to empirical analysis and does not provide a formal convergence analysis. Conversely, we have shown results by simulating a federated environment. Showcasing it in real-world applications on low-resource devices, such as mobile phones, would expedite implementation. The applicability of our method beyond language tasks, such as vision transformers is untested.

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

## A  MAC Operations

The layerwise operations and their corresponding OPs count are detailed in Table 6. The main components are computing query (Q), keys (K) and values (V), attention layers followed by fully connected layers. For each of the PEFT methods, we provide the components specific to them. In Table 7, we provide a summary of the number of layers (L), attention heads per layer (H), and the hidden dimension of the model (d) for all considered models.

| Layer | Fwd/Bwd | Computation | OPs |
|---|---|---|---|
| QKV | Fwd | $Y = WX$ | $L(Td^3)$ |
| | Bwd | $dX = dYW^T, dW = dYX$ | $L(Td^3 + Td^2)$ |
| Attention | Fwd | $Y = AV, A = SM(QK^T)$ | $L(T^2d^2 + T^3d)$ |
| | Bwd | $dV = dYA, dA = dYV, dQ = dAK, dK = dAQ$ | $L(T^2d^2 + T^3d + T^3d + T^3d)$ |
| FC | Fwd | $Y = WX$ | $L(Td^3)$ |
| | Bwd | $dX = dYW^T, dW = dYX$ | $L(Td^3 + Td^2)$ |
| LoRA | Fwd | $Y = BZ, Z = AX$ | $L(Td^2r + Tr^2d)$ |
| | Bwd | $dZ = dYB, dB = dYZ, dA = dZX, dX = dZA$ | $L(Td^2r + Trd + Tdr + Tr^2d + Td^3)$ |
| IA3 | Fwd | $Y = l_xX$ | $L(Td)$ |
| | Bwd | $dl_x = XdY, dX = l_xdY$ | $L(Td + Td)$ |
| Prompt Tuning | Fwd | $Y_p = WX_p, Y = AV, A = SM(QK^T)$ | $T_pd^3 + (T_p+T)^2d^2 + (T_p+T)^3d$ |
| | Bwd | $dX_p = dY_pW^T, dV = dYA, dA = dYV, dQ = dAK, dK = dAQ$ | $T_pd^3 + (T_p+T)^2d^2 + (T_p+T)^3d + (T_p+T)^3d + (T_p+T)^3d$ |
| P-Tuning | Fwd | $Y_p = WX_p, Y = AV, A = SM(QK^T)$ | $L(T_pd^3 + (T_p+T)^2d^2 + (T_p+T)^3d)$ |
| | Bwd | $dX_p = dY_pW^T, dV = dYA, dA = dYV, dQ = dAK, dK = dAQ$ | $L(T_pd^3 + (T_p+T)^2d^2 + (T_p+T)^3d + (T_p+T)^3d + (T_p+T)^3d)$ |

Table 6: Layerwise computations and their corresponding OPs. $X$: Input, $Y$: Output, $d$: hidden dimension, $T$: Number of tokens, $W$: Weights, $Z$: Intermediate output, $A$: Attention output, $A, B$: Low-rank approximation of $W$, $l_x$: Learnable scaling factors in IA3, $T_p$: Number of prompt tokens, $X_p$: Prompt token inputs, $Y_p$: Prompt token outputs, $L$: Number of layers.

| Model | L | H | d |
|---|---|---|---|
| T5-Small | 18 | 8 | 512 |
| T5-Base | 36 | 12 | 768 |
| BART | 36 | 16 | 1024 |
| DistilBERT | 12 | 12 | 768 |
| RoBERTa | 12 | 12 | 512 |
| GPT-2 Small | 12 | 12 | 768 |

Table 7: Model configurations: Number of layers (L), number of attention heads per layer (H), and hidden dimension (d).

