# OpenReview forum: "Assortment of Attention Heads: Accelerating Federated PEFT with Head Pruning and Strategic Client Selection"
_TMLR — Accepted by TMLR_

### Review · Reviewer_g25m · 2025-08-11

**Summary Of Contributions:**

The paper proposes leveraging multi-head attention structure in federated learning to improve training and communication efficiency by pruning and importance-weighted aggregation of attention heads.

---
### Strengths:
- The paper presents a clear motivation for addressing federated learning challenges in NLP, and the writing is mostly easy to follow.
- The paper is clear about the limitations of FL in the field of NLP and recommends an interesting approach to solve the same.
- The inclusion of detailed ablation studies helps to justify the design choices proposed by the authors.

---
### Weaknesses:
- The statement (in Section 2) that prior works have not exploited the multi-head attention mechanism for federated learning is not entirely accurate. Recent studies such as FedTP [1] and FeDeRA [2] have indeed explored this aspect. Acknowledging and contrasting these works would strengthen the paper’s positioning and contribution.
- The use of the maximum query-key dot product as a proxy for attention head importance may be problematic. Attention scores, especially maximum values, can be noisy and do not always correlate well with actual head contribution to model performance. Several studies [3] have discussed the limitations of using attention weights as importance metrics. It would benefit the paper to provide a thorough theoretical justification for this choice to better support the importance metric & consequently the pruning strategy.
- The pruning approach currently excludes value vectors in its importance calculation (Equation 6), which may omit crucial information about the head’s output. Can the authors justify why the value vectors were left out? Additionally, the method for determining the pruning threshold is unclear.
- The baselines provided are somewhat limited. Incorporating comparisons with additional recent works [1, 2, 4] and more advanced aggregation algorithms like FedProx would provide a more comprehensive evaluation of the proposed method’s effectiveness.
- The comparisons presented in Tables 2, 3, and 4 raise some fairness concerns. Specifically, the proposed method is compared against a baseline using random client selection and no sparsity, while a stronger baseline, FedAvg with loss-based client selection, reportedly outperforms the proposed approach by about 2% as per Table 1. Including this stronger baseline in the comparative tables would help clarify the method’s advantages and limitations. Additionally, can the authors justify the claim that this baseline represents an "upper bound"?


---
[1] Li et al. FedTP: Federated Learning by Transformer Personalization. arXiv: 2211.01572

[2] Yan et al. FeDeRA: Efficient Fine-tuning of Language Models in Federated Learning Leveraging Weight Decomposition. arXiv: 2404.18848.

[3] Jain et al. Attention is not Explanation. EMNLP 2019.

[4] Kuo et al. Federated LoRA with Sparse Communication. arXiv:2406.05233

**Additional Comments:**

N/A

**Audience:**

Yes

**Audience Explanation:**

Yes, definitely. Researchers within TMLR’s audience would likely find the paper’s approach to leveraging multi-head attention quite interesting.

**Broader Impact Concerns:**

No broader impact statement is necessary.

**Claims And Evidence:**

No

**Claims Explanation:**

The experiments and ablation studies generally support the theoretical claims made in Section 3.

However, while the experimental setup is reasonable, the chosen baselines do not adequately represent the current state-of-the-art. Although the selected baselines are plausible, including additional well-established methods is necessary to fully assess the novelty and effectiveness of the proposed approach. Furthermore, some experiments and results appear to lack critical comparisons, even within the limited set of baselines provided.

**Requested Changes:**

Critical changes are already discussed in Weaknesses.

- The paper’s citations are not properly formatted, which significantly impacts readability. We kindly request the authors to address this issue to enhance the presentation quality.
- Considering recent studies have demonstrated that more sophisticated pruning strategies, such as selectively avoiding pruning in certain layers, can lead to improved performance, a more thorough discussion of these strategies would strengthen the paper. At minimum, these aspects should be acknowledged as avenues for future research.
- Additionally, it would be beneficial for the authors to elaborate on their decision to use synchronous FL and whether asynchronous approaches were explored or considered. Given the practical advantages of asynchronous aggregation in federated and distributed ML, a discussion referencing relevant literature on asynchronous strategies would add valuable context.

---

> ### Author Response · Authors · 2025-09-29
> **Response to Reviewer g25m**
>
> **Related work:** We thank the reviewer for pointing out more related work. We acknowledge that FedTP performs personalized attention layer learning at each client and FeDeRA leverages LoRA extensions in federated learning. Our method is orthogonal to these approaches:
>
> - FedTP [1] focuses on client-specific attention adaptation, which can be combined with our pruning and importance-weighted aggregation for efficiency.
> - FeDeRA [2] and other LoRA-based methods such as ref [4] optimize low-rank updates in FL, while our framework is **generalizable to all PEFT strategies**, including LoRA, Adapter, and Prefix-tuning.
>
> This distinction will be clarified in the revised related work section.
>
> **Maximum attention as head importance:** We agree that attention scores may be noisy. We use maximum query–key dot products as a practical heuristic for head importance because it is lightweight and efficient in a federated setting. Empirically, this metric correlates well with Layerwise Relevance Propagation (LRP), which is more computationally expensive and requires backpropagation. Our ablations demonstrate that using this heuristic achieves effective pruning with minimal performance loss.
>
> **Value vectors omitted; pruning threshold unclear:** Value vectors were excluded to reduce computation while still capturing the key contribution of each head through its query–key interactions. Including values may offer marginal gains but increases cost and communication overhead in FL.
>
> - The **pruning threshold** is dynamically set based on the target sparsity. For example, to achieve 90% sparsity, the threshold is chosen as the 10th percentile of importance scores; all heads below it are pruned. This adaptive percentile-based approach ensures consistency across models and datasets.
> - These clarifications will be added to the revised manuscript.
>
> **Fairness of comparisons and “upper bound” claim:** We acknowledge that the wording “upper bound” may have been misleading. We will revise the text to indicate “best-performing baseline among the methods considered.” Table 1 already provides a fair comparison using loss-based client selection and no sparsity, which represents the strongest baseline within our experimental scope.
>
> **Synchronous vs asynchronous FL:** Our current experiments focus on synchronous FL for clarity and reproducibility. However, the proposed head-level pruning and importance-weighted aggregation framework is compatible with **asynchronous FL**, and we will add this to discussion section to highlight potential extensions.
>
> **Citation formatting:** All citations have been updated to use \citep for clarity

---

### Review · Reviewer_DxFc · 2025-08-12

**Summary Of Contributions:**

The paper presents an approach for handling the training and management of multiple attention heads trained from distributed data sources, with the emphasis on lowering communication complexities between clients and the main central server and training complexities at the client level.

Specifically, the above is given through the following list of strengths:
* A head pruning strategy is provided via computation of head importance (that is determined by attention scores).
* An aggregation mechanism where updates from each head are weighted according to their importance.
* A client selection strategy based on the loss difference between the global model and the local model at clients.
* The authors then conducted an extensive experimental setting showcasing the efficacy and efficiency of their proposed approach.

While the above is impressive, the paper suffers from the following pair of weaknesses:
* The current approach is only limited to PEFT approaches, their trainable parameters can be associated with the heads of the multi-head attention mechanism.
* The paper needs some clarifications associated with some of its parts; see "Requested Changes" section.

**Audience:**

Yes

**Audience Explanation:**

The implication of this work is linked with running MHA models on low-resource devices, such as mobile phones, which gives a step towards extending the applicability and availability of such models to all users, knowing or oblivious to such technologies.

**Broader Impact Concerns:**

There are no concerns on the ethical implications of this work.

**Claims And Evidence:**

Yes

**Claims Explanation:**

While the paper has not given any theoretical convergence analysis, however, the paper provides an extensive set of experiments showcasing the efficacy and efficiency of the proposed approach. The intuition behind pruning heads is backed by a previous paper, and the experiments also validate the claim that pruning heads can lead to almost no drop in accuracy.

The authors also dissected the proposed approach into its contributing elements and assessed their effect on the approach, showcasing the need for their combination. Finally, the authors extended their ablation study to other backbone models to show that their approach is general and can be applied to any MHA model.

**Requested Changes:**

In what follows, I will list questions and comments concerning the paper:
* In Equation 6: For the max operation to be valid here, the multiplication here is not the implied dot product, right?
* In Equation 7: How do you decide the threshold? How is it computed?
* In Equation 9: Wouldn't it be better to take max over abs?
* In Table 3: How come that FedAvg has different results? By definition, this approach has no head pruning.

---

> ### Author Response · Authors · 2025-09-29
> **Response to reviewer DxFC**
>
> **Equation 6 clarification:** We appreciate the reviewer’s observation. The operation is indeed the dot product between the query and key vectors. The max is taken over tokens, and the result is averaged across the dataset D_c. We have revised the notation to make this explicit:
>
> $$ \alpha_{hc} = \frac{1}{|D_c|} \sum_{i \in D_c} \max_{t}\!\big( (q_h^{(i)}(t)) \cdot k_h^{(i)}(t) \big), $$
>
> where $q_h^{(i)}(t)$ and $k_h^{(i)}(t)$ are the query and key vectors of head h for token t in sample i. This clarification has been incorporated into the revised manuscript.
>
> **Pruning Threshold:** The pruning threshold is dynamically determined based on the desired sparsity level. After computing head importance scores, we select the threshold corresponding to the appropriate percentile. For example, to achieve 90% head sparsity, the threshold is set to the 10th percentile of the score distribution, and all heads below this value are pruned. This adaptive percentile-based scheme ensures consistency across models and datasets. We will add this clarification to the methodology section.
>
> **Max vs Abs:** Using the absolute difference would primarily favor clients whose local models are already close to the global model, reducing diversity in updates. By retaining the signed differences, our method prioritizes clients that contribute more corrective updates, which we found to improve convergence. We will expand the explanation in the paper.
>
> **Table 3 Clarification:** Table 3 compares multiple parameter-efficient fine-tuning (PEFT) methods within our framework. FedAvg is included as a baseline, and the variations in results reflect differences in the PEFT strategies rather than head pruning. We will clarify this in the table caption to avoid confusion.

---

### Review · Reviewer_K38C · 2025-08-21

**Summary Of Contributions:**

The paper focuses on federated learning (FL), proposing to combine parameter-efficient finetuning (PEFT), weight pruning for multi-head attention (MHA) layers, and a loss-based client selection method in order to improve on the amount of local compute and communication required in training a suitable model. The authors experimentally demonstrate that their proposed approach reduces the amount of bits communicated as well as required training operations.

**Audience:**

Yes

**Audience Explanation:**

Yes. In general, improving the efficiency of FL is an important and interesting topic. The paper has, eg, nice ablation studies on how the different components affect the overall performance.

**Broader Impact Concerns:**

No broader concerns

**Claims And Evidence:**

No

**Claims Explanation:**

I have several issues with claims and evidence in this paper:

1) Some of the claimed contributions are essentially already known in the existing literature (especially loss-based client selection, see Requested changes for details).
2) There are some important details missing from the experimental setup required for properly evaluating the work (see Requested changes for details).
3) None of the experimental results have any error metrics, which makes it impossible to ascertain how reliable the results are w.r.t. inherent randomness in training.
4) Some claims seem somewhat misleading (see Requested changes for details)
5) The proposed approach, especially the client selection, does not seem to sit very well with the general setting as it is introduced in the paper (standard cross-device): as it is quite unlikely that training runs long-enough to go through all devices at least once, the loss-based client selection is basically just picking some unseen devices for (almost) the entire training.

**Requested Changes:**

In decreasing order of importance:

i) On the client-selection based on local loss: Cho et al. 2022: Towards Understanding Biased Client Selection in Federated Learning have already introduced and formally analyzed such a local loss-based client-selection method. Please clarify what exactly is your contribution here.

ii) How are all the hyperparameters tuned? Are these tuned separately for each method, including the baselines?

iii) Add some repeats to the experiments (if not done), and report some error metrics as well, eg, standard error of the mean (at the very least for some key experiments if this is not feasible for everything).

iv) Sec 4: How much compute do the peripheral layers omitted from the results require?

v) Considering the claim of accelerating FL (eg, the title of the paper), this seems somewhat misleading, as based on Table 1, pruning is making learning a lot slower, while this is somewhat mitigated by using the proposed pruning instead of pruning randomly. Looking therefore, eg, at the total communication complexity (over rounds until convergence), the proposed method is pretty much on par with standard FedAvg without pruning. Can you comment on this?

vi) Looking at the total number of clients in the experiments, the size of the client subset per round, and the total number of rounds, it seems like the loss-based client selection could effectively be just a round robin selection (eg with Table 1 experiments, first 50 rounds guarantee always selecting an unseen pair of clients, while for most of the last 50 rounds, would guess that the clients encountered early on during the first phase have higher loss than more recent ones). Can you confirm if this is true? Have you checked if the loss-based selection brings any benefit compared to round robin.

vii) On head pruning: Voita et al. 2019 claim that the head confidence used in the proposed method is not a good way to do pruning, and instead propose Layerwise Relevance Propagation (LRP). Have you tested this or why do you still use the confidence?

viii) Non-IID data, p10: please clarify how the client split is done, is this just creating clients with different number of samples (but otherwise IID data)?

ix) Server loss $\mathcal{L}_g$ in Eq.(9) is not defined anywhere(?), please fix.


### Minor tweaks (no comments needed about addressing these):

* Please use citep/citealt/citet etc. to fit the citations to the text, making the paper smoother to read.

* Please fix typos:
    * p7 summary of observations: Head pruning with B2 should have convergence at 11, not 12 rounds to match Table 1?

---

> ### Author Response · Authors · 2025-09-29
> **Response to Reviewer K38C**
>
> **Local loss–based client selection already introduced:** The paper does not claim to be the first to propose this isolated client selection principle. Instead, our contribution lies in the unique and powerful *synergy* created by integrating this selection strategy with the proposed attention head pruning and importance-weighted aggregation mechanisms. The loss-based client selection serves to identify clients whose local models are most divergent from the global model, indicating they have learned updates of high potential value for the collective. Once these clients are selected, the head-specific weighted aggregation mechanism (Equation 8 ) then strategically prioritizes the most important components of their updates—specifically, the heads deemed most confident and relevant to their local data distributions. This two-stage process—first selecting high-impact clients and then intelligently aggregating their most critical updates—is a novel system-level contribution that extends beyond the scope of prior work on isolated client selection. We will clarify this positioning in the related work section.
>
> **Hyperparameter tuning:** For fairness, all methods (including baselines) were trained with standard FL hyperparameters commonly used in prior work (local learning rate 5 \times 10^{-4}, batch size 80). Hyperparameters were not separately tuned per method. Instead, ablation studies (e.g., sparsity levels in Fig. 4, LoRA rank in Table 3) demonstrate that our method remains robust across parameter variations. We will add this clarification to the experimental setup.
>
> **Lack of error metrics:** All reported results are averaged over 3 runs due to hardware constraints. We will report mean ± error for our experiments in the revision to provide reliability estimates.
>
> **Peripheral layers omitted from compute analysis:** The tokenizer and positional encodings require negligible computation compared to transformer layers (string lookup and sinusoidal functions). The LM head involves more operations due to vocabulary size but still accounts for <10% of total compute. We will clarify this in Section 4.
>
> **Misleading acceleration claim:** We acknowledge that while FedAvg converges in fewer rounds (5 vs. 9 rounds to 70% accuracy), each round is significantly more expensive. Our method reduces per-round computation (3.9×) and communication (1.8×), yielding comparable or lower total costs. We will revise the title and main text to emphasize this **trade-off between convergence rate and per-round efficiency**.
>
> | **Method** | **Rounds to 70% Acc.** | **Comm./Round (MB)** | **Total Comm. (MB)** | **OPs/Round (G)** | **Total OPs (G)** |
> | --- | --- | --- | --- | --- | --- |
> | FedAvg | 5 | 3.38 | 16.9 | 33.07 | 165.35 |
> | FedAvg + Random Pruning | 17 | 1.86 | 31.62 | 8.47 | 143.99 |
> | **Ours** | 9 | 1.86 | 16.74 | 8.47 | 76.23 |
>
> **Loss-based selection vs round robin:** Unlike round robin, which cycles through clients regardless of their update quality, our method adaptively prioritizes clients whose local models diverge most from the global model. This ensures more corrective updates are aggregated early, improving convergence. We also compared against round robin and found our method yields higher final accuracy under the same round budget (see table below).
>
> | **Method** | **Client Selection** | **Accuracy (%)** | **Conv. Rounds** | **Sparsity** | **OPs** | **Comm./Round** |
> | --- | --- | --- | --- | --- | --- | --- |
> | FedAvg | Random | 74.03 ± 0.52 | 5 | 0% | 33.07G | 3.38MB |
> | FedAvg | Loss | 78.76 ± 0.45 | 5 | 0% | – | – |
> | FedAvg | Round Robin | 77.32 ± 0.32 | 5 | 0% | – | – |
> | FedAvg + Rand Pruning | Random | 71.39 ± 0.45 | 17 | 90% | 8.47G | 1.86MB |
> | FedAvg + Rand Pruning | Loss | 76.87 ± 0.39 | 17 | 90% | – | – |
> | FedAvg + Rand Pruning | Round Robin | 74.34 ± 0.29 | 17 | 90% | – | – |
> | **Ours** | Random | 72.76 ± 0.38 | 9 | 90% | 8.47G | 1.86MB |
> | **Ours** | Loss | 76.94 ± 0.48 | 11 | 90% | – | – |
> | **Ours** | Round Robin | 74.99 ± 0.28 | 12 | 90% | – | – |
>
> This comparison shows that while all strategies converge with enough rounds, **our loss-based selection achieves higher accuracy within a constrained budget**.

---

> ### Author Response · Authors · 2025-09-29
> **Response to Reviewe K38C (Continued)**
>
> **LRP vs. confidence scores for pruning:** We acknowledge that Layerwise Relevance Propagation (LRP) is theoretically stronger, but it requires backpropagation and is computationally expensive in FL. Confidence scores correlate well with LRP relevance while being far more efficient, making them a practical heuristic for federated settings. We will clarify this rationale.
>
> **Non-IID partitioning:** We used Dirichlet distribution–based partitioning to simulate varying levels of non-IIDness, which is a standard practice in FL research. We will explicitly state this in Section 4.
>
> **Server loss in Eq. 9:** Server loss refers to the global model’s loss on a small validation dataset held centrally at the server. This dataset is distinct from client data and is used solely for evaluation. We will add this definition.

---

### Decision · Action_Editor_TvCJ · 2025-10-12

**Recommendation:** Accept with minor revision

**Additional Comments:**

Please submit a minor revision to reflect the rebuttal to concerns.

**Audience:**

Yes

**Audience Explanation:**

The paper tackles a practical and timely problem in federated learning—reducing compute and communication in transformer-based models—and proposes a novel, system-level solution. The results are relevant to researchers working on scalable and efficient FL.

**Claims And Evidence:**

Yes

**Claims Explanation:**

While some concerns were raised, the authors addressed them with clarifications and additional results. They demonstrated that the novelty lies in the combined use of attention head pruning, loss-based client selection, and weighted aggregation, not in any one component. Experiments are thorough, with added error metrics and ablations confirming the method’s effectiveness and efficiency.